# Stability and Requirement for Thiamin in a Cell Culture Feed Used to Produce New Biological Entities

**DOI:** 10.3390/cells12020334

**Published:** 2023-01-16

**Authors:** Alisa Schnellbächer, Aline Zimmer

**Affiliations:** Upstream R&D, Merck KGaA, Frankfurter Straße 250, 64293 Darmstadt, Germany

**Keywords:** thiamin, cell culture media, metabolomics, stability, LC-MS, biomanufacturing

## Abstract

Thiamin is susceptible to heat and oxidation, which is a concern for the development of concentrated and room temperature stable feeds used to produce recombinant proteins. Hence, it is critical to understand the reactivity and necessity of the vitamin in liquid feeds to be able to either develop mitigation strategies to stabilize the vitamin or to remove thiamin from formulations if it is unnecessary. LC-MS/MS was used to investigate thiamin stability in different liquid feed formulations and to identify thiamin degradation products. Results indicate oxidation of thiamin and interaction with amino acids, keto acids, and sulfur containing components. Thiamin necessity in feed was assessed during a fed batch experiment, focusing on cell performance and critical quality attributes of the produced recombinant proteins. The impact of thiamin depletion in the feed on the intra- and extracellular metabolome was investigated using untargeted LC-MS/MS. Results indicate that thiamin can be removed from the feed without affecting the performance or the intra- and extracellular metabolome of the tested cell lines. Overall, profound insights on thiamin reactivity and necessity are presented in this study, suggesting the removal of the dispensable and instable vitamin as a simple means for the development of next generation feeds used to produce therapeutic biological entities.

## 1. Introduction

Cell culture media are developed to support the growth and maintenance of mammalian cells in vitro as well as the production of new biological entities such as mAbs [1]. Whereas older formulations are supplemented with undefined ingredients such as hydrolysates or serum, newly developed media are completely chemically defined [1]. These formulations regulate osmotic pressure and pH with salts and buffers, and consist of energy sources such as glucose, large amounts of amino acids (AAs) as well as small quantities of trace elements and vitamins [1]. Such complex cell culture media mixtures may be subjected to stress such as light exposure or increased temperature during preparation or storage, which might result in the degradation and interaction of nutrients, possibly negatively impacting cellular performance and recombinant protein quality [2,3,4,5]. This reactivity is an even greater issue for cell culture feeds used in fed batch (FB) processes due to a higher component concentration [6]. For instance, when exposed to light, riboflavin induces photooxidation of Trp (tryptophan) or folic acid, leading to the production of reactive oxygen species such as hydrogen peroxide [7], which might further oxidize media components. Increased formation of Trp oxidation products may not only change the color of the feed [6], but it might also affect the color of the produced therapeutic protein [3], which is a critical quality attribute (cQA).

Mitigation strategies for unstable feeds often include storage at 4 °C, protection from light, addition of stabilizers, or preparation of backbone formulations that are supplemented prior to use with stock solutions of unstable components [8]. While these approaches might be practical at a small scale, storage at 4 °C of large volumes for industrial biomanufacturing is expensive and preparation of several concentrated solutions is time-consuming [8]. Ideally, concentrated feeds that can be stored at room temperature and are composed of a minimum of ingredients are required. To achieve this goal, a deep understanding of the reactivity and necessity of cell culture media and feed components is critical.

In this study, thiamin, an essential vitamin for the cellular energy metabolism, was studied. Thiamin consists of a pyrimidine and a thiazole ring linked by a methylene bridge, and is phosphorylated inside the cell at the primary hydroxyl group to its active form thiamin diphosphate (TDP) (i.e., cocarboxylase) [9]. In addition to the diphosphate, mono-(TMP) and triphosphate (TTP) species also exist, the biological roles of which still remain unknown [10]. 

TDP is a functional coenzyme able to catalyze the oxidative decarboxylation of 2-keto acids via pyruvate dehydrogenase (PDH), α-ketoglutarate (KG) dehydrogenase (OGDH), branched-chain α-keto acid dehydrogenase (BCKDH), and 2-oxoadipic acid dehydrogenase (OADH) in order to fuel the tricarboxylic acid cycle. Furthermore, thiamin is necessary for the activity of transketolase, an enzyme critical for the pentose phosphate pathway [11,12,13].

In this work, thiamin was demonstrated to be instable in a liquid feed formulation when stored for three months at room temperature. Thiamin instability was investigated using LC-MS technology to understand the major degradation routes as well as the interaction partners in a solution. Further, necessity of the vitamin in a feed formulation was evaluated during a FB process to evaluate whether the vitamin may be removed from future formulations. Cell performance parameters (growth, viability, and protein production), cQAs (glycosylation, aggregation, and charge variants) of the produced recombinant proteins were monitored to evaluate the requirement for the vitamin in feeds. In addition, extra- and intracellular metabolomic analyses were performed using untargeted LC-MS/MS to investigate the impact of the vitamin removal from the feed on metabolic pathways.

## 2. Materials and Methods

### 2.1. Reagents

Raw materials and cell culture feeds were all purchased from Merck, Darmstadt, Germany, if not stated otherwise. Chemical standards for LC-MS/MS feature identification were purchased from commercial suppliers. Details of the origins of the standards are provided in the Appendix A. The feed formulations 1, 2, and 3 used in this study were customized dry powder derived from Cellvento^®^ Feed220 or Cellvento^®^ ModiFeed Prime, depleted in AAs, vitamins, or other organic compounds, and were reconstituted according to the manufacturer’s guidelines, with compounds added back as desired. 

### 2.2. Stability Studies in Feed

Samples of feed at pH 7.0 ± 0.3 (50 mL) were sterile-filtered through a 0.22 µm polyethersulfone filter and stored in 50 mL centrifuge light-protected tubes (Merck, Darmstadt, Germany) for a maximum of 98 days at 23 °C or 37 °C or 15 days at 70 °C. Samples were taken under sterile conditions and were frozen at −20 °C for subsequent LC-MS/MS analysis. 

### 2.3. LC-MS Feature Determination and Structure Elucidation

Time point series from stability study experiments were analyzed untargeted using UPLC (Vanquish, Thermo Fisher, Waltham, MA, USA) coupled with an electrospray-ionization quadrupole time-of-flight (ESI-Q-TOF) mass spectrometer (Impact II, Bruker, Billerica, MA, USA) [6]. Briefly, the samples were diluted 10 times in water prior to LC-MS/MS analysis. Five microliters of sample were loaded in 99.9% buffer A (20 mM ammonium formate/0.1% FA) onto a XSelect HSS T3 column (2.1 × 150 mm, 3.5 µm, Waters, Milford, MA, USA) thermostated at 40 °C with a flowrate of 300 µL/min. An optimized 12 min linear gradient was applied using 100% methanol (buffer B) as follows (minute/% B): 0/0.1, 2/0.1, 4/20, 6/30, 8/80, 8.5/100, 9.5/100, 9.6/0.1, 12/0.1.

LC-MS/MS analyses were performed in triplicate using the Impact II mass spectrometer equipped with an ESI source (Bruker, Bremen, Germany). MS acquisition was performed in positive and negative modes with capillary voltages set at 4500 V and 3500 V, respectively, and the end plate offset set at 500 V. Nebulizer and dry gas (250 °C) were set at 1.4 bar and 9.0 L/min, respectively. MS spectra were acquired over the *m*/*z* range 20–1000 with a scan rate of 12 Hz followed by data-dependent auto-MS/MS acquisitions using a fixed MS-MS/MS cycle time of 0.5 s and a summation time adjusted to the precursor intensity.

Data analysis was performed using Data Analysis 4.0 (Bruker) and Progenesis QI (Non-linear Dynamics—Waters). Briefly, LC-MS/MS raw data were processed using the automatic peak detection algorithm of Progenesis QI (retention time (RT) between 0.8 min and 11.0 min, adducts: M+, M + H, M + Na, M + K, 2M + H, M + 2H). Unique ions (RT and *m*/*z* pairs e.g., 1.23_456.7890 *m*/*z*) were grouped, and their abundances were summed to generate unique features. No normalization was applied to the dataset due to the nature of the study (more features created during storage compared to the starting material).

An untargeted search was applied in stored thiamin-depleted and thiamin-containing feed to find thiamin degradation and interaction products with the following filter criteria: fold change > 2; corrected ANOVA q-value < 0.05; hierarchical clustering to select features that were generated from thiamin; common thiamin fragments in MS/MS spectrum if MS/MS data were available (thiamin MS/MS data in Appendix A). These selection parameters were used to generate a database of thiamin-derived features. Feature annotations were performed within Progenesis QI using precursor mass, fragment mass, and RT (if available) with tolerances set at 5 ppm, 15 ppm, and 0.3 min, respectively. When features could not be identified with the used databases, a sum formula was calculated (used settings: elements H, C, N, O, S, P; precursor tolerance 3 ppm; isotope similarity 95%) and a putative structure was proposed. Features were curated manually for in-source fragments and wrongly assigned adduct/isotope features. Details regarding the LC-MS tier assignments of features according to Sumner et al. [14] and some further published guidance document [15] are provided in the Appendix A. 

A targeted search was applied using the “thiamin-derived feature” database in compound specific deficient feeds, e.g., feed without riboflavin, to identify thiamin interaction partners. A feature was considered as a degradation product when the feature was formed in every condition except the thiamin-deficient condition. If the feature was also absent in a component-specific deficient feed condition, the feature was likely generated from the interaction between thiamin and the removed compound. 

A targeted search was applied using the “thiamin-derived feature” database in stored feeds 1, 2, and 3. For the representation as heatmap, samples were normalized on the area under the curve (AUC) of a stable and non-saturated feature present in each study (valine: 1.76_118.0868 *m*/*z*).

### 2.4. Fed-Batch Cultivation in Spin Tubes

Cell culture experiments with four different Chinese hamster ovary (CHO) clones producing three distinct mAbs and a fusion protein were performed with four biological replicates in spin tubes with vented caps (TPP, Trasadingen, Switzerland) at 37 °C, 5% CO_2_, 80% relative humidity, and an agitation speed of 320 rpm. CHO-K1 GS, referred to as clone 1, CHOZN GS -/-, referred to as clone 2, clone 3, and clone 4 producing mAb1, mAb2, mAb3, and a fusion protein were cultivated at a starting density of 0.2 × 10^6^ cells/mL in 30 mL Cellvento^®^ 4CHO media at pH 7.0 (Merck, Darmstadt, Germany). Cells were fed either with feed 3 containing CKG (**3**; 2-(2-carboxyethyl)-1,3-thiazolidine-2,4-dicarboxylic acid) at pH 7.0 ± 0.3 containing or lacking thiamin. The following feeding strategy was applied at *v*/*v* ratios: 3% on day 3, 4.5% on day 5, 6% on day 7 and day 10, and 2.5% on day 13. The glucose concentration was fed on demand up to 6 g/L during the week and was increased up to 13 g/L before the weekend. The cell performance (viable cell density, viability) was assessed daily with the Vi-CELL^TM^ XR 2.04 cell counter (Beckman Coulter, Fullerton, CA, USA). Glucose and IgG concentrations were monitored daily with the bioprocess analyzer CEDEX BIO HT (Roche, Mannheim, Germany).

### 2.5. Antibody Purification and Critical Quality Attribute Analysis

Recombinant proteins were purified from the cell culture supernatant on day 12 of the FB experiment by using protein A PhyTips^®^ (PhyNexus Inc., San Jose, CA, USA).

Aggregation profile was analyzed using size exclusion chromatography coupled to an UV detector (SEC-UV) using an Acquity UPLC (Waters, Milford, MA, USA) and a TSK gel SuperSW series column (Tosoh Bioscience, Griesheim, Germany) at room temperature. Further, 10 μL sample, adjusted to 1 mg/mL with storage buffer (85% (*v*/*v*) of 30 mM citric acid pH 3.0 and 15% (*v*/*v*) of 0.375 M Tris Base pH 9.0), was applied to the system at a flow rate of 0.35 mL/min. In addition, 0.05 M sodium phosphate and 0.4 M sodium perchlorate solution adjusted to pH 6.3 were used as mobile phase [16]. 

Glycosylation patterns were analyzed with LC-MS using the GlycoWorks^TM^ RapiFluor-MS^TM^ N-Glycan Kit (Waters, Milford, MA, USA). Briefly, purified IgG were deglycosylated and labeled according to the manufacturer guide. The released and labeled glycans were analyzed using UPLC with an ACQUITY UPLC Glycan BEH Amide Column (300 Å, 1.7 µm, 2.1 × 150 mm^2^) coupled to an ACQUITY UPLC^®^ Fluorescence (FLR) Detector (Ex: 265 nm and Em: 425 nm). Glycans were characterized by their mass-to-charge ratio in the MS (Synapt G1 HDMS; Waters) with an ESI source in positive mode. The scan time was set to 1 min, and the mass range was 100–2250 Da with the following settings: 2.5 kV capillary, 30 V sample cone, 3 V extraction cone, 100 °C source temperature, 350 °C desolvation temperature, 50 L/h cone gas, and 750 L/h desolvation gas. The acceptable mass error of the system was ±20 ppm. The flow rate of the UPLC was set to 0.5 mL/min with an injection volume of 18 µL and a column temperature of 45 °C. Two solvents were used: 50 mM ammonium formate (pH 4.4) and acetonitrile with a gradient of 55 min (0 min 20:80, 3 min 27:73, 35 min 37:63, 36.5 min 100:0, 39.5 min 100:0, 43.1 min 20:80, 47.6 min 20:80, and 5 min 20:80) [17].

Distribution of charge variants was determined with capillary isoelectric focusing (cIEF) using the CESI8000 (Sciex, Framingham, MA, USA) according to the supplier’s protocol. 

### 2.6. Metabolomics

#### 2.6.1. Extracellular Metabolomics

Supernatants from the FB experiment were collected daily and stored at −80 °C for subsequent LC-MS/MS analysis. Supernatants were diluted 1:10 with water prior to LC-MS/MS analysis.

#### 2.6.2. Intracellular Metabolomics

Cell suspension from days 3–17 (2 mL; 500 µL from each biological replicate) was immediately quenched with 8 mL of 0.9% NaCl at 0.5 °C. Cells were centrifuged (2500 rpm, 1 min, 4 °C), the supernatant was discarded, and the pellet was snap-frozen in liquid nitrogen, and stored at −80 °C. To extract intracellular metabolites, pellets were incubated 15 min on ice, extracted by vortexing 1 min in pre-chilled buffer (−20 °C; 40% acetonitrile, 40% methanol) to yield 5 × 10^7^ cells/mL, and further incubated in a thermomixer (1600 rpm, 15 min, 1 °C). Samples were centrifuged at 18,000× *g* for 15 min at 1 °C. The supernatant was dried using a speedvac (45 °C at 45 min, 100 mTor, 55 min total run time). Dry samples were stored at −80 °C. Prior to analysis with LC-MS/MS, pellets were reconstituted (equivalent of 5 × 10^7^ cells/mL) in pre-chilled water. 

#### 2.6.3. LC-MS/MS

LC-MS/MS analysis was performed similarly as described in 2.3. Intracellular metabolomic samples were normalized on the protein concentration quantified using a Bradford assay.

### 2.7. Cytotoxicity Assay

Potential toxicity of found thiamin degradation products was monitored using the CellTiter-Glo^TM^ Luminescent Viability Assay (Promega, Madison, WI, USA) as described elsewhere [6]. 

### 2.8. Statistics

Statistical and graphic analyses, as well as calculation of the AUC, were performed with Prism 9.1.2 software (GraphPad Software Inc., La Jolla, CA, USA) or Tableau 2021.1.10 (Tableau Software, Seattle, WA, USA).

## 3. Results

### 3.1. Thiamin Stability

#### 3.1.1. Thiamin Stability in Cell Culture Feed

In order to investigate the stability of a thiamin-containing feed when stored long-term (3 months) at increased temperatures up to 37 °C, stability studies were performed in feed 1; samples were collected at regular intervals and analyzed with LC-MS/MS. During these experiments, Trp, His, and Met were identified as unstable AAs and a detailed investigation of Trp and its degradation product formation was carried out and published elsewhere [6]. Focusing next on the stability of vitamins, thiamin (**1**) was identified as the most unstable vitamin when stored at elevated temperatures (Figure 1A,B). In solution, thiamin (**1**) is sensitive to oxidation and reduction with degradation rates depending on pH and temperature, resulting in structural changes at the highlighted groups as presented in Figure 1C [18,19,20,21,22,23]. Investigations of thiamin (**1**) stability in further formulations containing different cysteine (Cys) sources (S-sulfocysteine (SSC, **2**) in feed 2 or CKG (**3**) formed by Cys and KG in solution in feed 3) revealed a decreased stability of the vitamin when stored at 37 °C (Figure 1D,E), which was hypothesized to result from interactions with sulfur containing molecules or with the keto acid KG in case of the CKG (**3**) containing condition.

#### 3.1.2. Thiamin Degradation and Interaction Product Identification

In order to understand how thiamin degrades in complex feed formulations, feed with or without the vitamin was stored for 9 days at 70 °C to simulate a fast degradation. Stored thiamin-depleted feed was compared to the vitamin-containing control condition using untargeted LC-MS/MS. LC-MS features were characterized with unique identifying information (RT and *m*/*z* for a given parent ion for example 1.23_456.7891 *m*/*z*). To find thiamin-derived products, the analysis was focused on signals significantly differing between both conditions and whose MS/MS data contained thiamin-specific fragments e.g., 122.0714 *m*/*z* (if MS/MS data were available; Appendix A for MS/MS data of thiamin (**1**) and 2.3 for detailed data processing description). 

As a next step, an internal database including already-known thiamin degradation products was used to identify interesting features. Unfortunately, only a few features (23 features) were assigned with a tier 1 level according to best practices for small molecule structure identification (Appendix A for tier level 1–5 explanation). 

To ease the identification of the remaining unknown thiamin-derived products, a strategy was applied to find out whether the feature was a thiamin degradation or interaction product and, if it was the latter one, the respective interaction partner. Hence, specific component-deficient feed formulations e.g., feed—Phe or feed—Glu, were stored for 15 days at 70 °C and compared to a control condition containing all feed ingredients using targeted LC-MS/MS. A feature was reported as a degradation product when the feature was generated in every condition except the thiamin-deficient one. In contrast, if a feature was absent in a specific component-deficient condition, the feature was hypothesized to result from the interaction between thiamin and the lacking constituent. For example, 6.56_287.1504 *m*/*z* was not detected in the conditions without thiamin and Phe, indicating that the feature was, very likely, an interaction product between both (Figure 2A,B; only data for AA shown since no other component class interacted with thiamin (**1**); Appendix A for data of remaining features). 

In order to analyze thiamin degradation in the presence of the Cys sources SSC (**2**) or CKG (**3**), a similar approach was used after inclusion of the sulfur-containing compounds. For example, 5.17_363.0789 *m*/*z* was only generated when thiamin (**1**) and SSC (**2**) were present, suggesting that this feature was an interaction product between both components (Figure 2C and MS/MS data of in-source fragment 5.12_283.1221 *m*/*z* in Figure 2D as 5.17_363.0789 *m*/*z* was fragmented in-source, leading to the direct release of the labile -SO_3_ group).

This approach and literature knowledge were essential to suggest possible structures based on the analysis of the MS/MS data. When possible, standards were purchased to obtain a confirmation for the proposed structure.

The structure proposals for each feature were assigned a tier level based on best practices for small molecule structure identification. From the 74 features detected as possible thiamin degradation/interaction products, 23 features were assigned as tier 1 and 37 as tier 3, corresponding to a total of 20 confirmed compound structures and 30 putative annotations as some features were identified as isomers (Figure 2E–H; absolute configuration unknown, features with earlier RT were designated as “a” and later RT as “b”). Remaining features in this experiment were classified as tier 4 and 5 and their corresponding abundance data can be found in Appendix A.

Found tier 1–3 degradation and interaction products can be grouped according to their type of degradation or their interaction partner: (1) interaction products with sulfur-containing species; (2) interaction products with keto acid KG; (3) interaction products with AAs; (4) oxidation products (Figure 2E–H).

#### 3.1.3. Quantification of Formed Degradation Products in Stored Feed

After thiamin degradation products were found and annotated/identified, their formation was investigated in a real time stability study of maximum 98 days in feed with or without the addition of the respective Cys sources stored at elevated temperatures.

For interesting features with an abundance of minimum 10,000 at both temperatures in any feed, the integral over time of the abundance was plotted for each feature and the relative percentage of each AUC was calculated compared to the highest AUC value of the respective feature. An increase in intensity is presented as color change from light to dark blue in Figure 3.

In feed 1 without additives stored at 23 °C, mainly oxidation products were detected such as thiochrome (**41**), (oxidized) parts of thiamin (**43**, **45**, **48**, **49**, **53**) as well as an interaction product with Phe (**36**) or di- and trisulfides (**7**, **8**, **22**, **23**). These signals were mostly increased when the feed was stored at 37 °C, even though a few *m*/*z* (e.g., *m*/*z* corresponding to **53** and **41**) showed a lower abundance at higher temperature, indicating a further degradation of the molecule.

When SSC (**2**) was added in feed 2 at 23 °C storage, oxidation products of thiamin and parts thereof were only slightly decreased, while di- and trisulfide formation as observed in feed 1 was nearly completely inhibited. Instead, a disulfide with Cys (**14**) was formed, suggesting a thiol– or disulfide–disulfide exchange reaction between thiamin and SSC (**2**). At 37 °C storage, a disulfide between thiamin and the sulfite moiety of SSC (**10**) was significantly increased, possibly suggesting a decomposition of SSC (**2**) at that temperature. 

When CKG (**3**) was added in feed 3, generation of features detected in feed 1 was nearly completely inhibited at 23 °C storage, probably due to the antioxidative capacity of KG or CKG (**3**) [24]. However, supplementation of the keto acid also resulted in the formation of interaction products between KG and thiamin (especially **28**, which may be formed by reactions similar to those described between dehydrogenases and keto acids [25]). Storage at elevated temperatures led to the formation of further interaction products with KG (**25**–**27**) and Cys (**14** and **15**), as free Cys was probably available due to dissociation of CKG (**3**).

#### 3.1.4. Toxicity of Thiamin-Derived Degradation Products in CHO Cells

Since thiamin degradation products may be present during the FB cultivation process, their toxicity was investigated in a CHO K1 GS cell line. Compounds for which a standard was commercially available or synthesized on demand were dissolved in DMSO and then spiked into a CHO cell culture to achieve final compound concentrations of up to 1 mM. None of the tested compounds exhibited toxicity (data not shown).

### 3.2. Thiamin Necessity

Due to the high sensitivity of thiamin (**1**) to oxidation and the high reactivity with other cell culture feed components, the stabilization of the vitamin appears challenging. An alternative approach for feeds would be to eliminate thiamin from the formulation and rely exclusively on the vitamin levels provided by the medium. However, this requires a careful evaluation of the necessity for the vitamin in multiple processes and with highly demanding cell lines. 

Hence, the impact of thiamin removal from the feed was investigated during a FB process with four different cell lines. Tested parameters were cell performance (growth, viability, and protein production) and cQAs of the produced recombinant proteins (glycosylation, aggregation, and charge variants). TDP formation, as well as the overall impact of thiamin depletion in the feed on the intra- and extracellular metabolome, was investigated using untargeted LC-MS/MS. 

#### 3.2.1. Cell Performance and cQA Analysis

In order to evaluate the necessity of thiamin in feed, cell performance and cQAs of the recombinant protein produced by each cell line treated with thiamin-free feed were compared individually to their respective control condition. Charge, aggregation, and glycosylation profiles were determined with cIEF, SEC-UV, and UPLC-MS, respectively, with purified recombinant proteins from day 12.

Results presented in Figure 4A indicate a comparable performance for each clone independently of the presence of thiamin in the feed with a maximum viable cell density of 16 and 17 × 10^6^ cells/mL for clone 1 and 2, respectively. Even though the vitamin was removed from the feed, viabilities were maintained similarly for each clone above 75% until day 12 for clone 1 and above 90% until day 17 for clone 2 (Figure 4B). For both processes using either thiamin-containing or thiamin-free feed, final titers of roughly 4 and 7 g/L for clones 1 and 2 were reached (Figure 4C), indicating that the cell performance was not affected by thiamin removal from the feed. Results for two other cell lines are presented in Appendix A and support this conclusion.

Similarly to the cell performance, cQAs were not affected by the utilization of a thiamin-deficient feed. For instance, as presented in Figure 4D, a comparable aggregation profile was observed with an intact main peak on day 12 above 90% and 65% for mAb1 and mAb2, respectively. Charge variants of recombinant proteins produced with both processes were also similar, for instance, with 8.8% and 8.4% basic, 60.1% and 59.1% main, and 31.0% as well as 32.4% acidic species for mAb1 produced in clone 1 fed with the thiamin-deficient or the thiamin-containing feed respectively on day 12 (Figure 4E). Investigation of the glycosylation profile showed no difference for each cell line, independently of the used feed, e.g., on day 12 with 84.6% and 82.8% terminal N-acetylglucosamine (GlcNAc) species as the predominant form for mAb1 produced in clone 1 fed with thiamin-deficient or thiamin-containing feed respectively (Figure 4F). Results for two other cell lines are shown in Appendix A and show that thiamin was not an essential component in the feed; thus, it may be removed from feed formulations.

#### 3.2.2. Thiamin Phosphate Species

Since TDP is an important coenzyme for the activity of dehydrogenases involved in energy metabolism, its availability during the process was evaluated after removal of its precursor thiamin from the feed. Because thiamin and TDP can be also formed from TMP and TTP, respectively, these species were monitored as well using LC-MS/MS (Figure 5 showing clone 1 and 2; results for clone 3 and 4 in Appendix A).

While thiamin was nearly depleted extracellularly in all cell lines when the vitamin was removed from the feed, TDP was detected intracellularly during the whole experiment or until the viability started to decrease, indicating that the amount of thiamin in the basal was sufficient to support metabolic requirements. This conclusion was further supported by the intracellular presence of TMP and thiamin (with an abundance > 50,000 at any time during the FB), which may be converted into TDP if necessary.

Interestingly, while TTP was not detected in any condition, the nucleotide analogue adenosine thiamin triphosphate was identified intracellularly as tier 3 (mono- and diphosphate versions not detected). The biological role of this triphosphate remains unknown but was previously detected upon metabolic stress in bacteria [26]. 

#### 3.2.3. Extracellular Metabolomics

In order to investigate whether thiamin removal from the feed resulted in further differences in the CHO metabolome other than the decreased formation of phosphorylated thiamin species, supernatant was collected from the FB experiment and each cell line treated with thiamin-free feed was compared individually to its respective control condition using untargeted LC-MS/MS.

Within the selection criteria (according to Section 2.3; fold change > 2; corrected ANOVA q-value < 0.05), to find features that were differentially formed in the thiamin-depleted condition compared to the control, MS data were curated manually and features were annotated. 

For interesting features, the integral over time of the abundance was plotted for each cell line in both conditions and the relative percentage of the thiamin-free feed AUC was calculated compared to the control (Figure 6A for two cell lines, Appendix A for other cell lines). Nearly all features whose formation was reduced when the vitamin was removed from the feed were identified as tier 1 or tier 3 thiamin-derived compounds. Features can be grouped into putative non-enzymatically formed thiamin oxidation products (**41** and **53**) or interaction products with Cys (**14**) and KG (**26**) or enzymatic intermediate products from PDH (**57**), OGDH (**28**, can also be generated non-enzymatically), BCKDH or OADH (**60** and **61**), transketolase (**56**), or TMP (**54**) (Figure 6A), suggesting that only few specific pathways were impacted by the removal of the vitamin from the feed. 

#### 3.2.4. Intracellular Metabolomics

To confirm the extracellular findings, a similar strategy as the one employed in Section 3.2.3 was applied to compare the intracellular content of CHO cells when treated either with thiamin-containing or thiamin-depleted feed using untargeted LC-MS/MS.

Again, for the interesting features, the integral over time of the abundance was plotted in both conditions for each cell line and the relative percentage of the thiamin-free feed AUC was calculated compared to the control. According to the applied filter criteria (after 2.3; fold change > 2; corrected ANOVA q-value < 0.05; hierarchical clustering to select features that were thiamin-specific), many more intracellular features were considered interesting compared to the extracellular matrix, and just a fraction of these intracellularly found features was identified as tier 1 or tier 3. The top 18 features with an abundance > 100,000 were evaluated and are presented for two different cell lines in Figure 6B (Appendix A for other cell lines and remaining features with abundance < 100,000; putative structures for not-yet-presented tier 3 features in Appendix A). 

Presented features were categorized according to the already established groups for the extracellular dataset. The first group comprised thiamin-derived non-enzymatically formed degradation products (**43** and **48**). The second group contained enzymatically generated products from thiamin pyrophosphokinase (**55**), PDH (**57**–**59**), transketolase (**56**), OGDH (**28**, can also be generated non-enzymatically), and OADH or BCKDH (**60** and **61**). The respective mono- and di-phosphorylated species of transketolase, OADH or BCKDH intermediates were also considered interesting but with an abundance < 100,000. A third group was identified and consisted in features possibly not derived directly from thiamin e.g., inosine (**62**, tier 1) or (5-(6-amino-9H-purin-9-yl)-3,4-dihydroxytetrahydrofuran-2-yl)methyl dihydrogen phosphate (**63**, tier 3). 

Taking all criteria into account (cell performance, cQAs, TDP availability, impact on metabolome), thiamin concentration provided in the basal medium seems to be sufficient to supply four different cell lines with enough thiamin throughout the FB process. Since performance and metabolome were not impacted negatively by the absence of the vitamin from the feed, thiamin may be removed from feed formulations.

## 4. Discussion

In the first part of this study, thiamin stability was investigated in order to understand its degradation pathway and interaction with other components in different feed formulations. The second part investigated the necessity of thiamin in a specific feed formulation. 

Thiamin degradation and interaction with other compounds of the stored feed was shown to yield several products, the bioavailability of which remains mostly unclear even though the biological activity of some of these derivatives has already been studied. Indeed, modification of the reactive thiamin moieties (4-aminopyrimidine, methylene bridge, thiazole, and thiazole side chain) results in biological inactive products or thiamin antagonists that were shown in the literature to inhibit thiamin phosphorylation or TDP dependent enzymes [27]. For instance, cleavage of the methylene bridge, yielding for example **43** and **45**, was reported to result in complete loss of thiamin function in CHO cells, whereas toxopyrimidine (**44**) is known as a pyridoxine antagonist [28]. TDP-dependent enzymes can be inhibited by a substitution of the thiamin thiazole C-2 (binding site of the carbonyl substrate in enzymatic reaction; modified thiamin can still be phosphorylated), for example, when the 4-amino group of the pyrimidine ring is bound to the thiazole C-2 in case of thiochrome (**41**) or the thiazole C-2 is oxidized (**47**) [25]. In contrast, substitution of the thiazole C-2 with a decarboxylated keto acid such as **28** might be bioactive, as shown for the decarboxylated interaction product between thiamin and pyruvate [25]. TDP-dependent enzymes can also be inhibited by a modification of the 4-amino group of the pyrimidine ring that is important for the formation of an ylide (positive charge on the thiazole nitrogen and negative charge on the thiazole C-2 needed to attack carbonyl substrate in enzymatic reaction), for example, when the amino group is replaced by a ketone in the case of oxythiamin (**42**) [29,30]. The phosphorylation of thiamin is required for a tight binding of TDP to the apoenzyme and can be inhibited by a modification of the thiazole side chain, as observed with the putative product **50** [29,31]. While thiamin cleavage and oxidation products may result in a functional loss and enzyme inhibition, formed disulfides such as **22** and **7** might still be biologically active, as demonstrated for the thiamin analogs allithiamin, thiamin propyl disulfide, or thiamin tetrahydrofurfuryl disulfide [9,10]. Altogether, thiamin cleavage was reported to result in complete functional loss, while oxidation of the vitamin seems to form thiamin antagonists that can either inhibit thiamin phosphorylation or TDP-dependent enzymes. Disulfide formation and interaction with keto acids at the thiazole C-2 might actually yield bioavailable products that may be tested for their stability in feed and activity in CHO cell culture compared to thiamin.

Thiamin removal from the tested feed was shown to result in comparable cell performance of four CHO cell lines (including high producing and high growing clones) and similar cQAs of the produced recombinant proteins compared to the vitamin-containing control conditions. These results suggest that thiamin in the basal is sufficient to ensure coenzyme supply inside the cell throughout the whole experiment. Since the in-house feed formulation might contain several other non-essential components, the strategy developed in this study may be applied to identify and possibly remove these compounds from future formulations. As testing of each component is highly labor-intensive, the approach may be first applied to selected compounds such as other vitamins and trace elements. Even though vitamins are essential as coenzymes for the survival of the cell, they are only needed in small amounts, which might be sufficiently provided in the basal medium as demonstrated for thiamin [32,33]. Despite their crucial role for the cellular metabolism, which has already been extensively reviewed, vitamins are known to be unstable when exposed to light, heat, and oxygen, resulting in the destruction of the vitamin and the formation of degradation products that are, to some extent, harmful to the culture [5,7,34]. As a result, cell culture feeds containing these components need to be protected from light and heat during storage. These disadvantages can be simply avoided by removing non-essential vitamins from the formulation. Similar to the vitamins, trace elements are added in very low concentrations and their presence in culture is mandatory for the maintenance of cells, but trace element concentration in the medium might be sufficient to supply cells for a whole FB experiment [34]. Unlike unstable vitamins, the disadvantage of most trace elements in cell culture is their ability to trigger degradation of other compounds when added in high concentrations (e.g., formation of reactive oxygen species catalyzed by iron or copper) as well as their impact on process robustness due to contamination of raw materials with trace element impurities [16,34,35,36,37]. 

In conclusion, to be able to develop concentrated feeds that can be stored at room temperature and are composed of a minimum of ingredients, it is critical to investigate the reactivity and necessity of cell culture feed components as demonstrated for thiamin in this study.

## Figures and Tables

**Figure 1 cells-12-00334-f001:**
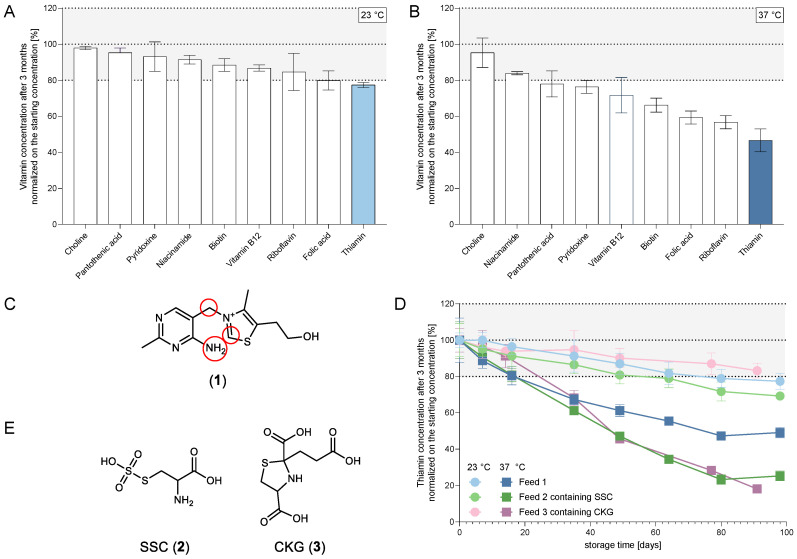
Vitamin concentration in feed 1 after 98 days storage at (**A**) 23 °C or (**B**) 37 °C normalized on the starting values. (**C**) Structure of thiamin (**1**) with chemical groups susceptible to oxidation and reduction highlighted in red. (**D**) Thiamin concentration in formulations containing different Cys sources (feed 2 containing SSC (**2**) and feed 3 containing CKG (**3**)) after maximum 98 days storage at 23 °C and 37 °C normalized on the starting values. (**E**) Structure of used Cys sources SSC (**2**) and CKG (**3**). The grey area represents the common technical variability of the method.

**Figure 2 cells-12-00334-f002:**
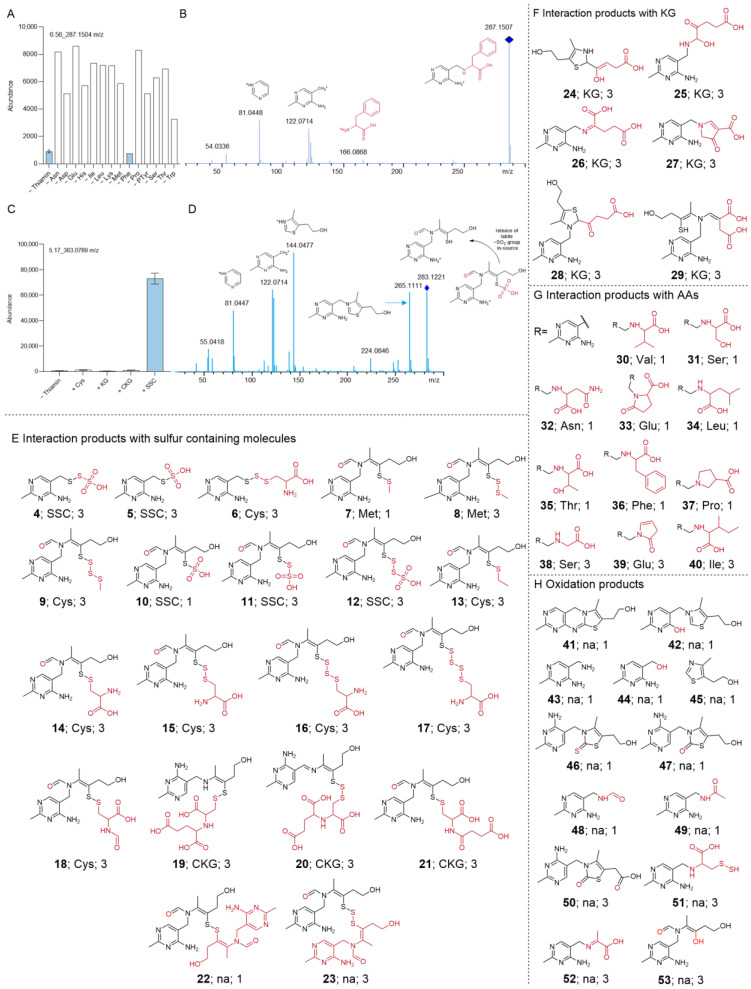
Strategies used to investigate thiamin instability and formed degradation/interaction products. (**A**) Example for identification of thiamin interaction partner via compound-deficient feeds, e.g., identification of Phe as the respective interaction partner of thiamin for 6.56_287.1504 *m*/*z* since the feature was not generated in thiamin- and Phe-deficient conditions. (**B**) Example of MS/MS data annotation, e.g., annotation of MS/MS data of 6.56_287.1504 *m*/*z* as interaction product between thiamin and Phe. (**C**) Example for identification of thiamin interaction partner via addition of supplements, e.g., identification of SSC (**2**) as the respective interaction partner of thiamin for 5.17_363.0789 *m*/*z* since the feature was not generated in thiamin- and SSC- deficient conditions. (**D**) Example for annotation of MS/MS data, e.g., annotation of MS/MS data of 5.12_283.1221 *m*/*z* (in-source fragment of 5.17_363.0789 *m*/*z*) as interaction product between thiamin and SSC (**2**). Structures are provided for tier 1 and tier 3 features that were identified as (**E**) interaction products with sulfur-containing species, (**F**) interaction products with keto acid KG, (**G**) interaction products with AAs, and (**H**) oxidation products. Component-specific information is presented as [**identifier**; interaction partner if identified, otherwise labeled as not available (na); tier level]. Red highlights structural differences compared to thiamin backbone.

**Figure 3 cells-12-00334-f003:**
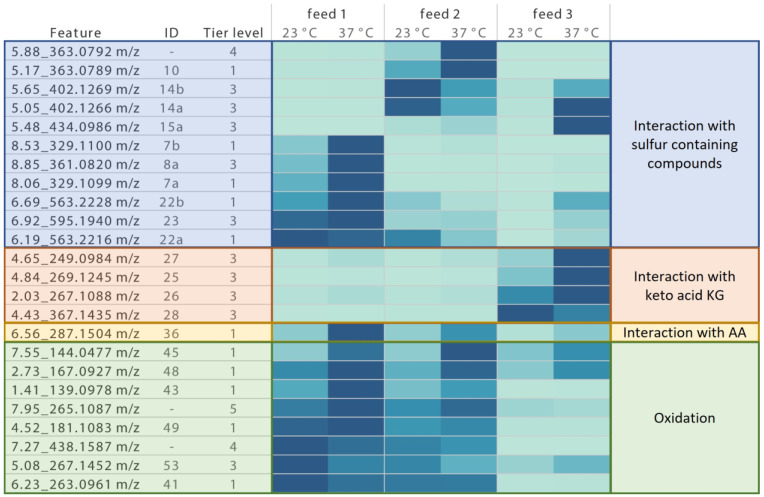
Impact of each additive on thiamin degradation product formation in liquid feeds when stored for 91 days or 98 days at 23 °C and 37 °C light protected. Only features with an abundance of minimum 10,000 at both temperatures were selected. AUC was normalized on highest value for each degradation product. Light blue color indicates that the feature was not detected or only background signal was detected. Change from light to dark blue indicates an increase in feature signal. Blue highlighted features indicate interaction products between thiamin- and sulfur-containing compounds. Orange highlighted features indicate interaction products between thiamin and KG. Yellow highlighted features indicate interaction products between thiamin and AAs. Green highlighted features indicate thiamin oxidation products.

**Figure 4 cells-12-00334-f004:**
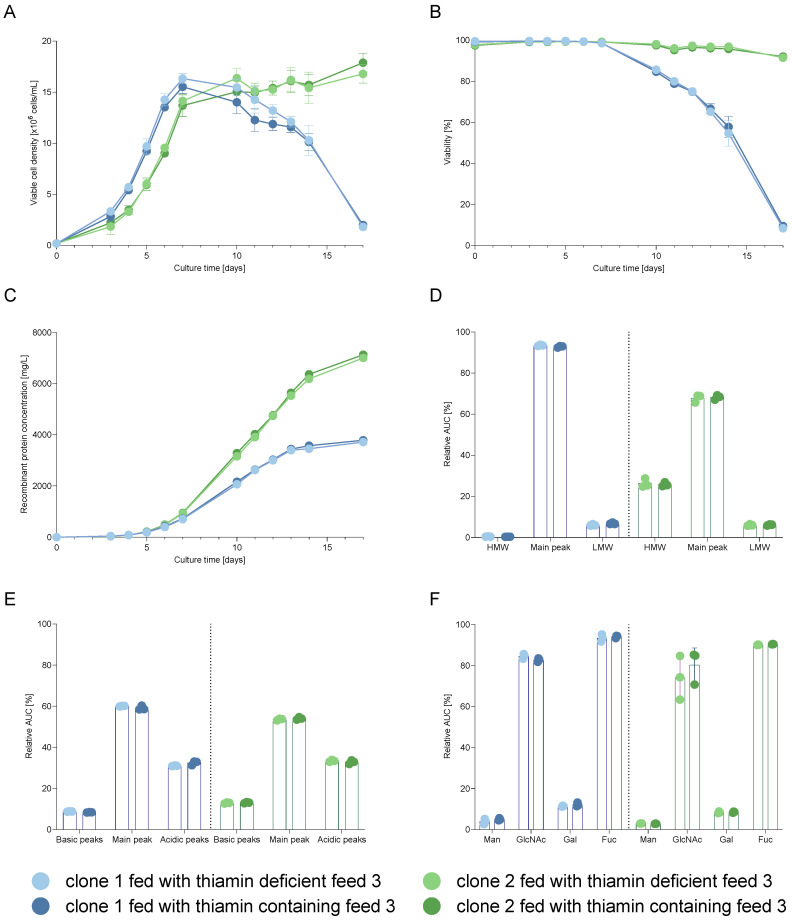
Cell performance of clone 1 and 2 during the FB and day 12 cQA analysis of mAb1 and mAb2. (**A**) Viable cell density, (**B**) viability, (**C**) titer, (**D**) relative quantification of intact recombinant protein and total low as well as high molecular weight forms (LMW, HMW), (**E**) relative quantification of charge variants, and (**F**) relative quantification of total fucosylation (Fuc), total high-mannose (Man), total galactosylation (Gal), and total N-acetylglucosamine (GlcNac) species. n_cell performance_ = 4 and n_cQAs_ = 3, error bars represent standard error of the mean.

**Figure 5 cells-12-00334-f005:**
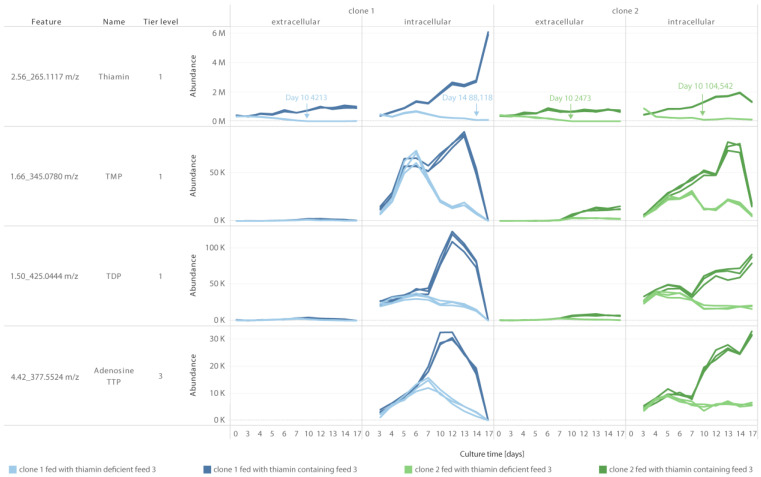
Thiamin species detected extra- and intracellularly during the FB. Lighter color indicates the conditions without thiamin in feed 3. Darker color indicates the conditions containing thiamin in feed 3. N = 3. Timepoint and lowest thiamin abundance highlighted for conditions fed with a thiamin-deficient feed.

**Figure 6 cells-12-00334-f006:**
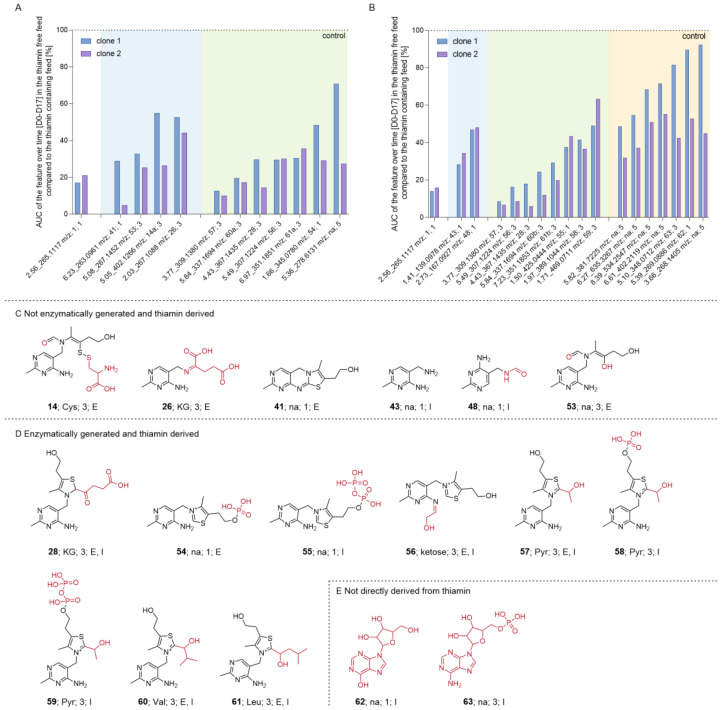
Extra- and intracellular metabolomics applied to supernatant or CHO cell lysates during a FB experiment. Area under the curve of (**A**) extra- or (**B**) intracellular interesting feature abundance over time (days 0–17) of thiamin-free feed conditions normalized to the control condition for clone 1 and 2. Tier 1- and 3-identified interesting features clustered in (**C**) not enzymatically or (**D**) enzymatically generated and thiamin-derived and (**E**) not thiamin-derived groups. For **60** and **61**, Val and Leu were presented as interaction partner. Component specific information are presented as [**Identifier**; interaction partner if found or not available (na); tier level; detected extra- (**E**) or intracellularly (I)]. Red highlights structural differences compared to thiamin.

## Data Availability

The data presented in this study are available on request from the corresponding author. The data are not publicly available due to reasons of privacy.

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
