# Peer review of "Stability and Requirement for Thiamin in a Cell Culture Feed Used to Produce New Biological Entities"

_cells, 2023, doi:10.3390/cells12020334_

Round 1
Reviewer 1 Report
This paper presents data on the degradation of thiamin under certain conditions and metabolites resulting therefrom. It also demonstrates that supplementary thiamin may not be necessary in culture media.
The paper is very well presented and the conclusions are appropriate based on the experimental data obtained. The study is detailed and extensive and warrants publication in Cells.
Author Response
Dear Reviewer,
We would like to thank you for taking the time and effort necessary to review the manuscript. We sincerely appreciate your valuable feedback.
Kind regards,
Aline Zimmer
Reviewer 2 Report
1. Fig. 5 has bad quality. Replace it
2. Please provide the choice of chromatographic conditions.
3. Clearly describe the practical significance of the work
4. Rewrite abstact
